# Identifying novel inhibitors against drug-resistant mutant CYP-51 *Candida albicans*: A computational study to combat fungal infections

Saadia Jabeen[1,☯], Muhammad Umer Khan[ID][1,☯], Hasan Ejaz[ID][2,*], Shakeel Waqar[3], Aisha Farhana[2], Muharib Alruwaili[2], Yasir Alruwaili[2,4], Abualgasim Elgaili Abdalla[2], Sahar Mudassar[5], Qurban Ali[ID][6,*]

1 Institute of Molecular Biology and Biotechnology, The University of Lahore, Lahore, Pakistan,
2 Department of Clinical Laboratory Sciences, College of Applied Medical Sciences, Jouf University, Sakaka, Saudi Arabia, 3 Department of Pathology and Laboratory Medicine, Auckland City Hospital, Auckland, New Zealand, 4 Sustainable Development Research and Innovation Center, Deanship of Graduate Studies and Scientific Research, Jouf University, Sakaka, Saudi Arabia, 5 Department of Pathology, Rashid Latif Medical College, Lahore, Pakistan, 6 Department of Plant Breeding and Genetics, Faculty of Agricultural Sciences, University of the Punjab, Lahore, Pakistan

☯ These authors contributed equally to this work.
* hetariq@ju.edu.sa (HE); saim1692@gmail.com (QA)

## Abstract

*Candida albicans (C. albicans)* is an opportunistic pathogen in immunocompromised individuals and a normal inhabitant of the oral cavity, throat, gastrointestinal tract, and genitourinary system among health populations. Our study focused on identifying new inhibitors capable of binding to the mutant cytochrome P450 family 51 (CYP-51) protein and intended to be effective against resistant *C. albicans* infections. The pharmacophore ligand-based model was used for the virtual screening of compound libraries. Molecular docking was performed on Maestro, Schrodinger. ADMET analysis was performed to check drug-likeness properties. Density function theory (DFT) calculations, molecular dynamic (MD) simulation, and free binding energy (MMPBSA) were also calculated. For docking, six compounds were selected from 11,022 hits from PubChem libraries, which showed the best interaction with mutant CYP-51 and were identified by pharmacophore mapping performed with the Pharma IT tool. Each of the six compounds was docked into the active site of the mutant CYP-51 protein. Overall, CP-3 exhibited significant binding affinity (−10.70 kcal/mol) as well as, showed good ADMET characteristics such as drug-likeness, absorption, distribution, metabolism, excretion, and toxicity. The lead compound, CP-3, was further used for MD simulation to observe the dynamic behavior of the complex in the active site of the mutant CYP-51 protein. Computational studies indicated that CP-3 could be a useful antagonist for the mutant protein, CYP-51. This study used computational approaches to identify potential inhibitors of *C. albicans* by targeting CYP-51 for antifungal drug development. Further *invitro* and *in vivo* studies are needed to evaluate its pharmacokinetic properties and efficacy as a novel antifungal drug.

**Data availability statement:** All relevant data are within the paper and its Supporting Information files.

**Funding:** This research is funded by the Deanship of Graduate Studies and Scientific Research at Jouf University through the Fast-Track Research Funding Program.

**Competing interests:** The authors have declared that no competing interests exist.

## 1. Introduction

Fungal infections are the major cause of death, accounting for more than 90% of the fatalities due to infectious disease. Every year, about 1.3 million people are killed by fungal infections globally. *Candida* species are the major cause of mycotic diseases. With a mortality rate of 48%–75%, *Candida* species threaten the lives of more than 400,000 humans globally [1]. *Candida albicans (C. albicans)* is a normal commensal fungus mainly present in the human vagina, oral cavity, and gastrointestinal system. It coexists with other bacteria, as well as the immune system of its host [2]. In a healthy host, *C. albicans* lives and thrives as a commensal without causing harm [3]. *C. albicans* is a commensal colonist of normal hosts, adopted to coexist within its human habitat. This necessitates homeostatic adjustment between the host's immune system and commensal microbiota because over-colonization may ultimately result in the development of infections. The overgrowth of *C. albicans*, leading to the disease called candidiasis, which normally occurs because the host has either an impaired immune system or dysbiosis [4].

Only a limited number of classes of antifungal agents are currently available, and fungal resistance to specific classes and multi-drug resistant (MDR) strains greatly reduces therapeutic options for patients [5]. The resistance of *Candida* and *Aspergillus* to azoles, as well as their resistance to echinocandin and the MDR of some *Candida glabrata* isolates, poses a major clinical challenge [6]. Similarly, the global emergence of azole resistance in *Aspergillus fumigatus* (partly due to the use of azoles in agriculture) and the emergent threats such as MDR *Candida auris* (*C. auris*) are equally concerning trends. The drug resistance of fungal species and the mechanisms by which resistance develops have arisen spontaneously (although functional explanations are likely, they cannot be evolutionarily proven) and through the acquisition of resistance by strains that were initially susceptible [7]. Altered drug efficacy due to decreased cellular uptake or increased efflux and changes to target sites within the cell lead to the production of biofilm-like architecture. Biofilms possess barriers that restrict antibiotic access. While *C. auris* is intrinsically multidrug-resistant, other species acquire resistance to multiple classes of antifungal agents by stepwise selection of different drug-resistance mechanisms [8].

In addition, *C. albicans* and other *Candida* species can rapidly acquire antifungal resistance by several mechanisms that allow the fungus to grow well during therapy. It is crucial to understand these mechanisms so that ways to prevent resistance and treat infections can be developed [9]. One mechanism involves small molecules that block the further transport of compounds loaded into vesicles, which serve to increase the intravacuolar pH and function like fungal efflux pumps. The major efflux pumps responsible for antifungal resistance include ATP-binding cassette (ABC) transporters such as candida drug resistance genes/differential display candidate genes Complementary-Determining Regions (CDR) i-e CDR1 and CDR2 [10]. ABC transporters encode a superfamily of proteins that pump azoles from the cell via an azole-H + transporter. The multidrug resistance gene (MDR1) functions as an exporter pump that confers azole resistance, which is two-way via the major facilitator superfamily (MFS) transporter [11].

Mutations affecting the target decrease the binding of antifungals and thus their kill rate [12]. Most such mutations occur in several essential components of a common key pathway—ergosterol (ERG) biosynthesis—and negatively affect azole antifungals' binding affinity or contribute to increased resistance (via mutations on ERG11 lanosterol 14-alpha demethylase) of invasive isolates or both. *Candida* species form biofilm on mucosal tissue as well as medical devices [13]. Biologically active aggregates grow in media and form dense, complex communities. The extracellular matrix acts as a barrier around the fungal cells and protects them from host defenses such as phagocytosis of immune-related cell types. Blastospores are a rare, treatment-resistant subpopulation of cells within *Candida* biofilms [14]. They may carry mutations in the ergosterol biosynthesis pathway, which can result in cross-resistance to

azoles and/or polyenes. Target enzyme overexpression results in parasites that carry combined intrinsic azole resistance (due to an increase in the target enzyme production). The activation of stress-responsive pathways offers *Candida* cells a marked survival advantage in the context of antifungals and other mutations, e.g., those mediating different routes to the ergosterol Erg3 or targets initiating breaks from polyenes interacting with cell ergonomic Erg5 [15].

Heat shock proteins (HSP) assist protein folding and stress tolerance during exposure to antifungals. They act as a regulator of both cell wall integrity and the stress response; their inhibition increases susceptibility to antifungals. However, *Candida* species can develop antifungal resistance [16]. The antifungal resistance of *C. albicans* is varied and incorporates efflux pumps, target site alterations, biofilm formation changes, and modifications of the ergosterol pathway and stress response signaling pathways, alongside genetic mutations. Resistance is acquired through gene inactivation (e.g., FKS1 gain-of-function mutations) or overexpression of regulator genes, resulting in transcriptional activation of efflux pumps, as well as multiple drug resistance mechanisms. A comprehensive understanding of these mechanisms is necessary as they may provide useful insights into the prevention and treatment of *Candida* infections without further exacerbating antifungal resistance. New treatments will require not only additional antifungal agents but also alternative and combination therapies and biofilm-disrupting strategies for better diagnostics to rapidly identify resistance [17]. The azoles, fluconazole, itraconazole, voriconazole, posaconazole, and isavuconazole, and the polyene, nystatin, are the most frequently used antifungals against mucocutaneous and systemic candidiasis. These antifungals inhibit ergosterol synthesis in the fungal cell membrane. However, *Candida* species have developed resistance to many of these drugs. Amphotericin B, another polyene, is effective against fluconazole-resistant *Candida* strains and invasive candidiasis [18]. It is a broad-spectrum anti-fungal that binds to ergosterol on fungal cell membranes, leading to the leakage of cellular contents and then death. Despite its high efficacy, amphotericin B cannot be used for long-term maintenance because of major concerns about its nephrotoxicity. Therefore, many scientists are in search of novel non-azole CYP-51 inhibitors due to its serious concern to the medical fraternity and resistance. A medical research also showed the better efficacy of non-azole compounds as antifungal agents [19]. Moreover, a study performed against resistant CYP-51 enzyme of *C. albicans*, revealed that the compound 1, 2, 4-triazine and its derivatives showed more inhibitory potential against CYP-51 enzyme compared to conventional antifungal drug fluconazole [20]. Pursuing this approach, our study aimed to screen novel compounds for the treatment of mutant cytochrome P450 family 51 (CYP-51) resistant *C. albicans* through computational studies.

## 2. Materials and methods

### 2.1. Predicting the drug-likeness of compounds in chemical libraries

Posaconazole, chosen as a reference molecule (R*), is a co-crystal ligand (CCL) that binds with the mutant state of RpsA and may inhibit the activity of the desired protein. The Pharmit tool was utilized for investigating the drug-like features of the R* compound. With the selection of all seven required properties, we were able to create a pharmacophore. The pharmacophore created with the reference molecule was used to filter PubChem compound libraries via Pharm IT. We discovered several compounds that interacted more strongly with the target than with the reference chemical.

### 2.2. Protein retrieval and protein model preparation

The protein data bank (PDB) was used to retrieve the targeted CYP-51 protein structure. The CYP-51 protein with the PDB ID, 5FSA, was selected due to its high protein crystallography

indicators as it showed a resolution of < 2.5 Å [21]. The initial protein preparation was carried out using PyMol software [22]. The Swiss-PDB Viewer was also used to enhance the quality of the protein for further downstream analysis. Mutant protein stability was evaluated using I-mutant, CUPSAT, Mcsm, and DUET, as discussed previously [23]. The structure was validated with Ramachandran plot, ERRAT, VERIFY 3D, and PROCHECK as performed by Shan M.A et al. [24].

### 2.3. Ligand retrieval and library preparation

We selected six bioactive compounds [25], as ligands from the PubChem Library using the Pharm IT tool. The 3D structures of these compounds were obtained from the PubChem database (https://pubchem.ncbi.nlm.nih.gov/) in SDF format. The selected ligand structures were prepared using ChemDraw 2D and 3D tools. The ligand preparation files were uploaded into Discovery Studio Visualizer and the ligands' 3D structures were generated (Table 1).

### 2.4. Ligand preparation and molecular docking

The Ligprep module of Maestro was used to prepare the selected ligands for docking. The protein was retrieved through PDB and was also prepared using the protein preparation wizard. The position of the co-crystal ligand was identified, and GRID was generated at the same position with an inner and outer box size of 10 Å and 30 Å, respectively. The CCL was docked on the GRID, and the docking was validated by generating the root mean square deviation (RMSD). The Maestro standard precision mode was used to perform docking. Multiple dock poses were created by Maestro, which showed the best confirmation position of the ligands with the targeted mutated protein. The docked scores were carefully evaluated, and the best-docked complex was selected for further molecular simulation analysis.

### 2.5. Structural interaction fingerprinting analysis

SIFT analysis was carried out on Maestro 12.5 (Schrodinger 2020-3) to gain insight into the overall interaction of the ligands with the key residues of the binding cavity. The docked complexes were imported to the maestro workspace and duplicates were filtered out to retain the best docked poses of the selected compounds. Further, the interaction fingerprint analysis

**Table 1. List of novel inhibitors against *Candida albicans* assessed through PubChem along with the 2D and 3D (S1 Table: Supplementary Material) molecular conformations of the compounds prepared using ChemDraw Professional 16.0.**

| Codes | Novel inhibitor | SMILES |
|---|---|---|
| CP-1 | [6-[4-[4-[2-(aminomethyl)-1H-imidazol-5-yl]phenyl]phenyl]-1H-benzimidazol-2-yl]methanamine | C1=CC(=CC=C1C2=CC=C(C=C2)C3=CN=C(N3)CN)C4=C-C5=C(C=C4)N=C(N5)CN |
| CP-2 | 3,5-bis[4-(3-hydroxypropoxy)phenyl]phenol | C1=CC(=CC=C1C2=CC(=CC(=C2)O)C3=CC=C(C=C3)OCCCO)OCCCO |
| CP-3 | (4-[2-[4-[(2-aminoethylamino)methyl]phenyl]ethynyl]-N-[(2S)-3-amino-1-(hydroxyamino)-1-oxopropan-2-yl]benzamide;methane) | C.C1=CC(=CC=C1CNCCN)C#CC2=CC=C(C=C2)C(=O)N[C@@H](CN)C(=O)NO |
| CP-4 | 7-[(1E,4E)-5-(5-carbamimidoyl-1H-indol-2-yl)penta-1,4-dienyl]quinoline-2-carboximidamide | C1=CC(=CC2=C1C=CC(=N2)C(=N)N)/C=C/C/C=C/C3=CC4=C(N3)C=CC(=C4)C(=N)N |
| CP-5 | 3-[3-fluoro-4-[(E)-3-hydroxyprop-2-enoxy]phenyl]-5-[4-(2-hydroxyethoxy)phenyl]phenol | C1=CC(=CC=C1C2=CC(=CC(=C2)O)C3=CC(=C(C=C3)OC/C=C/O)F)OCCO |
| CP-6 | 2-[4-[3-fluoro-4-[4-(1-hydroxypropan-2-yloxy)phenyl]phenyl]phenoxy]propan-1-ol | CC(CO)OC1=CC=C(C=C1)C2=CC(=C(C=C2)C3=C-C=C(C=C3)OC(C)CO)F |
| R* (CCL) | 4-[4-[4-[4-[[(3R,5R)-5-(2,4-difluorophenyl)-5-(1,2,4-triazol-1-ylmethyl)oxolan-3-yl]methoxy]phenyl]piperazin-1-yl]phenyl]-2-[(2S,3S)-2-hydroxypentan-3-yl]-1,2,4-triazol-3-one | CC[C@@H]([C@H](C)O)N1C(=O)N(C=N1)C2=C-C=C(C=C2)N3CCN(CC3)C4=CC=C(C=C4)OC[C@H]5C[C@](OC5)(CN6C=NC=N6)C7=C(C=C(C=C7)F)F |

was conducted, resulting in portraying the interaction matrix which summarized the overall interactions between the ligands and mutated target protein CYP-51 [26].

## 2.6. Pharmacokinetic parameters

An online tool SwissADME (http://www.swissadme.ch) was used to forecast the drug-like characteristics of the chosen ligands. It is considered as the most effective computational tool for predicting the pharmacokinetic features of newly synthesized drugs [27]. Additionally, a web server pkCSM (http://biosig.unimelb.edu.au/pkcsm/prediction), was employed for the estimation of the pharmacokinetic characteristics of the internally produced drugs [28]. A toxicity study was performed using the ProTox II and pkCSM servers [28], which predict electronic characteristics, geometry, mutagenicity, carcinogenicity, hepatotoxicity, immunotoxicity, and cytotoxicity in that order. The values of tiny ligand molecules were ascertained and predicted using cheminformatics and machine learning techniques. Because these internet technologies have a large dataset of chemicals at their disposal, their predictions are regarded as valuable [29,30].

## 2.7. Density Functional Theory (DFT) studies

Density functional theory (DFT) calculations were performed after slight modification of a previously described protocol [31]. The Gaussian 06 package (Rev.E.01) was used with the default configuration; all calculations in the SVP basis set used the B3LYP function. The electronic structure of atoms and molecules can be effectively calculated based on this theory. We ascertained the optimized geometric parameters, molecular electrostatic potential (MESP), and frontier molecular orbitals i-e. highest occupied molecular orbital (HOMO) and lowest unoccupied molecular orbital (LUMO), and global reactivity descriptors. The checks were examined using GaussView 6.

## 2.8. Molecular dynamics (MD) simulations

Protein-bound chemical structures can be changed and stabilized using MD simulations. We used the technology, LEaP, to build the protein and confer it specific criteria (ff14SB amber parameters) to better understand how these structures behaved over time. In addition, we applied a broader set of guidelines to the ligand or attached molecule. To ensure that the proton-containing protein in the LEaP system was balanced, two chloride ions (2 Cl⁻) were added. With a margin distance of 9.0 Å, the complex was solvated using LEaP and SPCBOX. The solvated complex was saved in PDB format; the LEaP process produced a parameter file and coordinate file. The compounds or natural ligands were reduced three times in order to eliminate stearic effects. During the first minimization step, ions and solvated water were used to optimize the protein and chemical compounds. The pocket residues of protein were optimized. During the final step of minimization, the entire system was adjusted to relax the protein-ligand complex containing the whole protein. After completion of the minimization step, the system was placed in the heating module and the temperature was gradually raised from 0 to 300 K. The system was further stabilized by performing additional equation procedures at a temperature of 300 K. MD simulation was performed using an NPT ensemble at a temperature of 300 K and pressure of 1 atm. Finally, using the CPPTRAJ module of AMBER 17 software, several parameters, including root mean square deviation (RMSD), root mean square fluctuation (RMSF), and radius of gyration (Rg), were observed via the following MD simulation of one complex [32]. To investigate the dynamic stability and sampling pattern, RMSD and 2D-RMSD were determined for the ligands, protein pocket, and apo-protein.

### 2.9. Binding free energy calculations and energy decomposition analysis

Using screenshots of MD simulation, binding free energies were computed for each complex in order to investigate its structural and energetic properties. Molecular mechanics-based analysis involving MMPBSA/MMGBSA molecular mechanics modules integrated into the AMBER 17 program was used to examine the binding free energy. For MM/GBSA calculations, 1000 snapshots from the last 2-ns MD trajectories were used to compute complexes. The binding free energy of every molecular species, including ligand-receptor complexes, free receptors, and free ligands, was computed by subtracting the total free energy of the ligand–protein complex ($G_{com}$) from the total free energies of individual proteins ($G_{pro}$) and ligands ($G_{lig}$) [33] as follows:

$$\Delta G_{bind} = \Delta H - T\Delta S = \Delta G_{com} - \left(\Delta G_{CDK9} + \Delta G_{lig}\right).$$

The following formula was used to calculate the free energy of the protein, protein–ligand complex, and ligand in the aforementioned equation:

$$\Delta G = \Delta E_{MM} - T\Delta S + \Delta G_{sol}.$$

The molecular mechanics energy fragmentation was further studied as Van der Waal (vdW) energies (EvdW) and non-bonded electrostatic fragmentation was further studied as vdW energies (EvdW), non-bonded electrostatic energies (Eele), and the solvation free energy (Gsol) divided into nonpolar solvation contributions and polar solvation energies. Binding free energies were then broken down into distinct residues that contributed to the deactivation interaction to investigate the decomposition of the parameters of vdW (GvdW) and electrostatic (Gele), polar (Gele, sol), and nonpolar (Gnonpol, sol) energies. Based on the binding free energy snapshots, decomposing parameters were also analyzed. Additionally, per residue-free energy decomposition analysis was also conducted on AMBER software as reported previously [33].

## 3. Results

### 3.1. *In silico* investigation of pharmacophore-based virtual screening of drug-like features

Pharmit is a web server that employs an algorithm for energy minimization and screens drug libraries using a pharmacophore model or molecular shape. Pharmit helps to screen large drug databases based on molecular structure or pharmacophore properties. We used Pharm to screen the PubChem database with posaconazole as a reference ligand (R*).

The drug-likeness analysis of posaconazole, a co-crystal ligand binding to the mutant form of CYP-51 and potentially inhibiting its activity, was performed using the validated 3D model (Fig 1). Pharmacophore features with the required properties of the template drug were generated. The pharmacophore generated for the reference compound was finally used to screen Pubchem database compounds. We found that a few compounds bound more tightly to our target than to the reference drug. A pharmacophore query was generated and 11,022 hits were obtained from the PubChem database based on their fit. The hits consisted of five hydrogen bond acceptors and one hydrophobic region; combined, they corresponded to the total number of possible hit areas. These hits were subsequently energy-minimized to find their most stable conformers, based on the molecular shape of posaconazole. They were then docked into the binding pocket of mutated CYP-51 to generate energy-minimized hits.

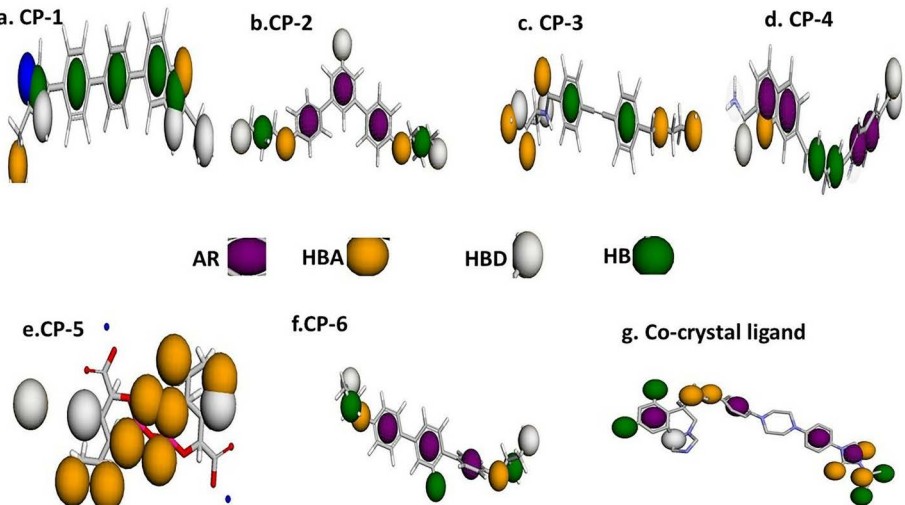

**Fig 1. Validated pharmacophore features of novel ligands with the reference co-crystal ligand R\* (posaconazole).**
(a) CP-1 [6-[4-[4-[2-(aminomethyl)-1H-imidazol-5-yl]phenyl]phenyl]-1H-benzimidazol-2-yl]methanamine (b) CP-2 (3,5-bis[4-(3-hydroxypropoxy)phenyl]phenol); (c) CP-3 (4-[2-[4-[(2-aminoethylamino)methyl]phenyl]ethynyl]-N-[(2S)-3-amino-1-(hydroxyamino)-1-oxopropan-2-yl]benzamide;methane); (d) CP-4 (7-[(1E,4E)-5-(5-carbamimidoyl-1H-indol-2-yl)penta-1,4-dienyl]quinoline-2-carboximidamide); (e) CP-5 (3-[3-fluoro-4-[(E)-3-hydroxyprop-2-enoxy]phenyl]-5-[4-(2-hydroxyethoxy)phenyl]phenol); (f) CP-6 (2-[4-[3-fluoro-4-[4-(1-hydroxypropan-2-yloxy) phenyl] phenyl] phenoxy]propan-1-ol); and (g) CP-R\* (the original co-crystal inhibitor).

## 3.2. *In silico* stability prediction of mutated protein structure-based analysis

In this study, CYP-51 sequences were predicted and missense mutations (Y132H), which showed beneficial outcomes, were retrieved using sequence-based predictors. Changes expected to destabilize protein structure (S1 Table) were tested using webservers I-Mutant (−2.19 kcal/mol), CUPSAT (−10.38 kcal/mol), mCSM (−1,702 kcal/mol), and DUET (DoG-SiteScorer; −1,597 kcal/mol).

**3.2.1. *In silico* validation of wild-type and mutated CYP-51 structure.** The computational approach was used to introduce the mutation into the wild-type protein of CYP-51 at chain A: Tyrosine-132 by replacing it with position A: Histidine (Fig 2a, 2b).

**3.2.2. Structural validation.** The Ramachandran plot (Fig 2c) showed that 94.6% of the residues of the mutated CYP-51 (Y132H) structure were present within the allowed regions, securing an ERRAT score of 91.1 and quality pass by VERIFY 3D and PROCHECK (S2 Table and Fig 2c).

## 3.3. *In silico* experimental validation of molecular docking of the mutated CYP-51 structure

Molecular docking is the most advanced approach for identifying protein-ligand interactions in molecular models. The molecular docking was performed on the top six compounds, which were screened out of 11022 compounds from ligands and compared to the reference R\* (CCL-posaconazole) of the target protein CYP-51. The spatial arrangement and structural organization of the target protein are detailed in Fig 3, revealing two identical subunits. This dimerization enhances the overall functionality by improving the catalytic activity of the protein as well as, the spatial arrangements revealing a globular structure of the protein.

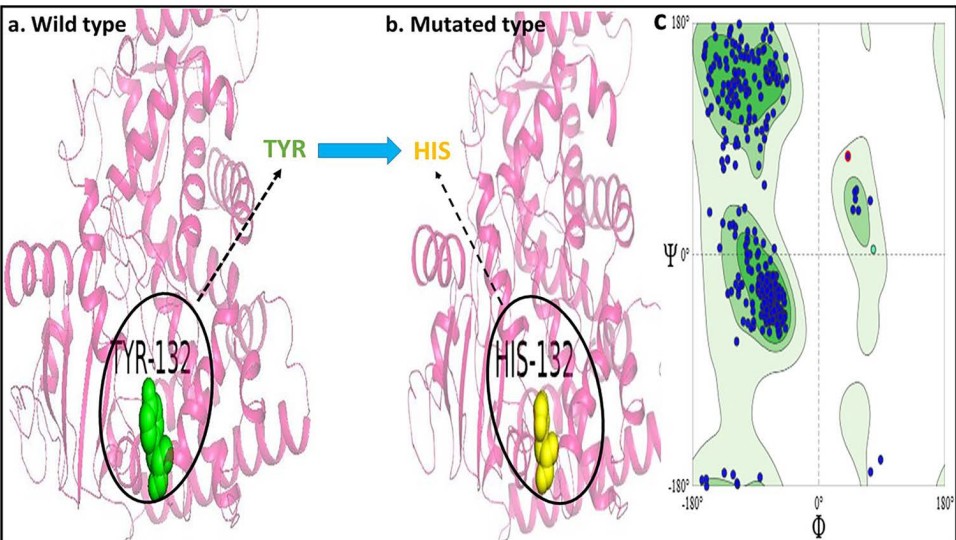

**Fig 2. (a)** Structural representation of the wild-type CYP51 protein, highlighting the tyrosine (Y132) residue in green; **(b)** Mutated CYP51 protein structure with the Y132H substitution, where histidine (H132) is depicted in yellow; **(c)** Validation of the mutated CYP51 protein structure through a Ramachandran plot, assessing the protein's stereochemical quality and conformational properties.

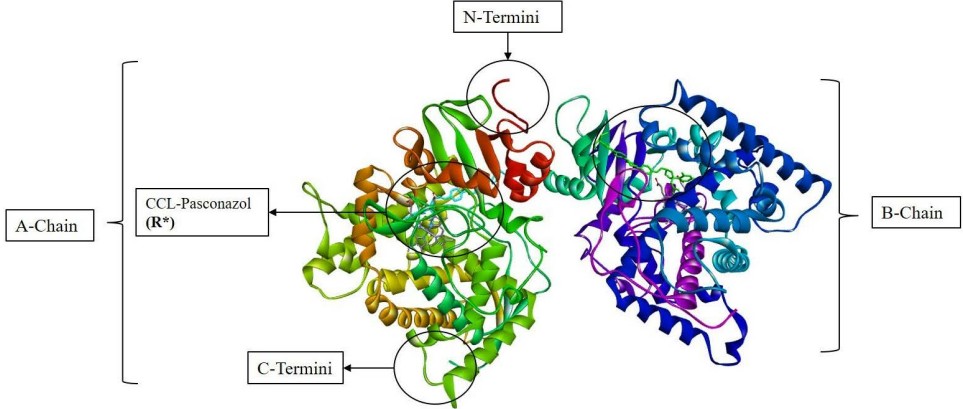

**Fig 3. 3D conformation of mutated target protein CYP-51, depicting the topological features of structure, visualized on Discovery Studio Visualizer.**

Additionally, the presence of a well-defined allosteric site indicated a suitable mode for ligand binding and interaction.

The docking scores illustrated in Table 2 revealed that CP-3 (−10.70 kcal/mol) scored the highest docking score among all six compounds followed by R* (posaconazole), achieving a score of −10.353 kcal/mol. These docking scores indicated a significant binding affinity against the mutant CYP-51 structure. These docking studies were validated by computing the RMSD of CCL by superimposing the re-docked CCL onto the original CCL, resulting in an RMSD value of less than 2Å.

The mode of binding interactions of the compounds against target protein CYP-51, in comparison to R* (CCL), was visualized as 2D and 3D and analyzed, shown in Figs 4 and 5,

**Table 2. Docking scores of top six natural compounds along with R\* compound, against mutated protein CYP-51 (PDB ID: 5FSA).**

| Codes | Docking scores (kcal/mol) |
|---|---|
| CP-1 | −9.952 |
| CP-2 | −10.021 |
| CP-3 | −10.70 |
| CP-4 | −8.715 |
| CP-5 | −8.237 |
| CP-6 | −9.576 |
| R\* (CCL) | −10.353 |

respectively. The R\* compound showed the formation of three conventional hydrogen bonds (H-bond) with the tyr-132 at a distance of 1.92Å, ser-378 at 2.41Å, and ala-61 at 2.89Å. These H-bonds were formed at an angle of 158.4°, 166.8°, and 177.47°, for each residue respectively. Additionally, a π-sulfur bond was also formed by the cys-470 residue with the heterocyclic amine of the R\* compound. These interactions indicated the stability and polarity of the compound within the allosteric site of the target protein. Moreover, non-covalent hydrophobic interactions including alkyl, π-alkyl, and carbon-hydrogen bonds were also visualized further stabilizing the hydrophobic pocket.

Subsequently, the docked complex of CP-1 – CYP-51 revealed three significant H-bonds with the $NH_3$ group of the compound. The following residues tyr-505 (2.04Å), his-377 (2.47Å), and ser-507 (1.77Å) formed a conventional H-bond at an angle of 132.5°, 161.01°, and 122.53°, respectively. Likewise to the R\* compound, a π-sulfur bond was also visualized, formed by the met-508 residue with the amide moiety of the compound. Furthermore, hydrophobic interactions were also visualized by neighboring residues of the binding pocket of mutated protein CYP-51. All these interactions significantly contributed to stabilizing the docked complex, indicating the polarity and hydrophobicity of the compound within the cavity.

Moreover, the CP-2 docked into the binding cavity of the mutated protein CYP-51, highlighted the formation of both polar and hydrophobic contacts. A total of three H-bonds were formed at various angles and distances with the carbonyl and hydroxyl groups of the compound. The following residues took part in H-bond formation tyr-64, ser-378, and phe-463 at a distance of 2.31Å, 2.64Å, and 2.63Å and an angle of 102.38°, 108.08°, and 113.29° for each respectively. Additionally, the hydrophobic contacts were also viewed by the met-508 and leu-376, forming non-covalent π-alkyl interactions, followed by the formation of carbon-hydrogen bonds and π-π stacking further dominating the hydrophobicity. These interactions revealed the polarity and stability of the complex.

Furthermore, the compound CP-3 formed strong binding interactions with the residues of the binding pocket. Both polar and hydrophobic contacts were formed by the residues of the mutated protein CYP-51 suggesting the stability of the compound within the hydrophobic pocket. The formation of conventional H-bonds was formed by the following six interacting residues: tyr-64, tyr-505, ser-378, his-377, gly-303, and ile-304 by the carbonyl and amine groups of the compound. The distance of the H-bond formed by each residue was 2.02Å for tyr-64, 1.80Å for tyr-505, 2.05Å for his-377, 2.22Å for ser-378, 2.93Å for gly-303, and 2.42Å for ile-304 at specific angles of 152.11°, 141.99°, 161.05°, 129.60°, 112.80° and 137.06° for each residue, respectively. Focusing on the H-bond distance and angles revealed that all interactions were formed within proximity of the compound. These contacts indicated the polarity of the docked complex. On the other hand, the hydrophobic interactions were formed by phe-233, leu-121, and leu-376 within the aromatic moiety of the compound along with the

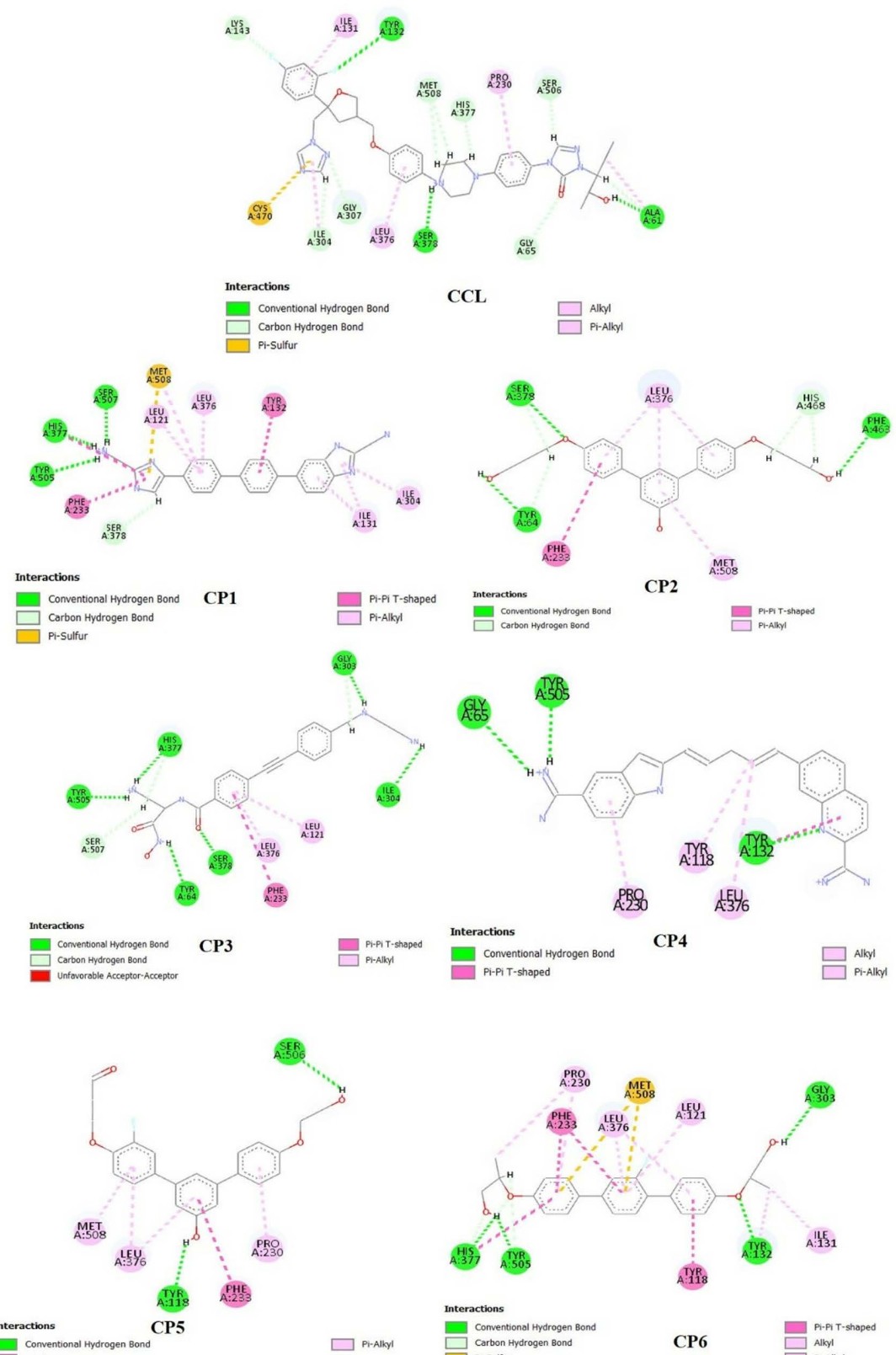

**Fig 4. 2-dimensional view of binding interactions of top six compounds, along with R\* (CCL) within the binding cavity of the mutated target protein CYP-51.**

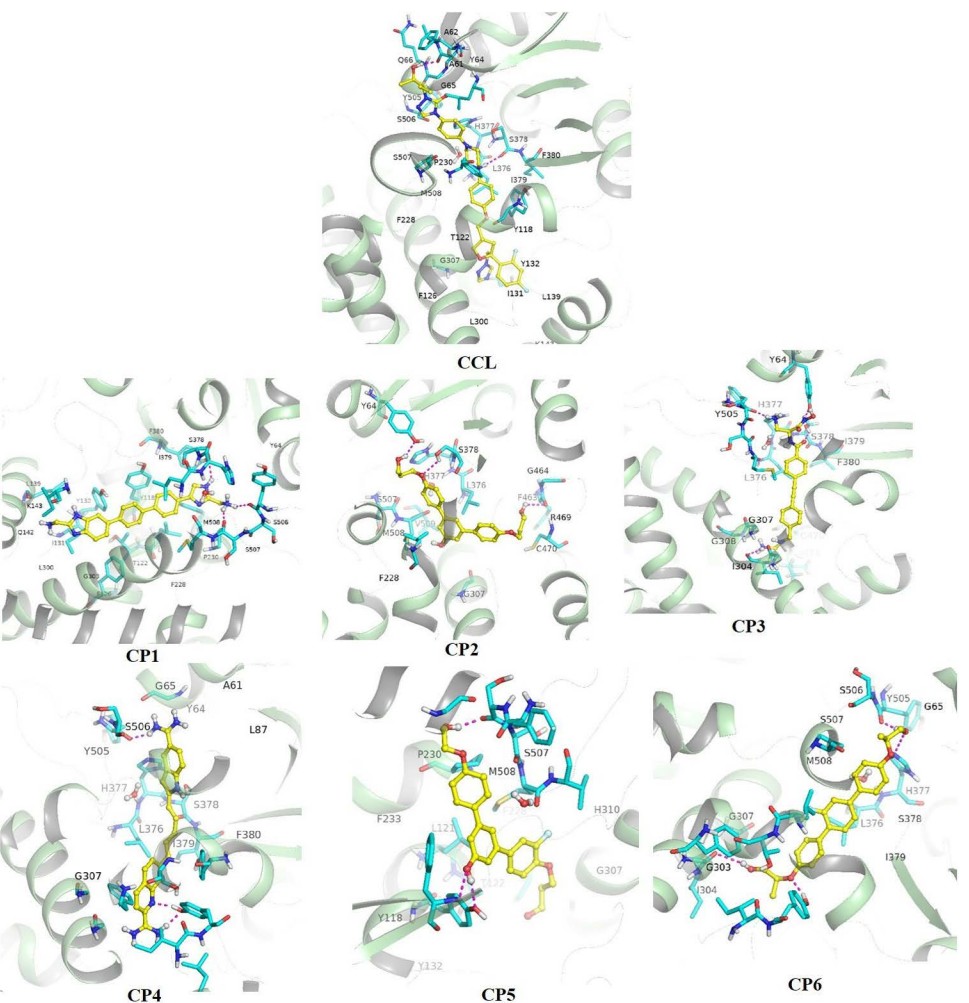

**Fig 5.  3-dimensional view of binding interactions of top six compounds, along with R\* (CCL) within the binding cavity of the mutated target protein CYP-51.**

formation of a carbon-hydrogen bond by ser-507 residue, contributing in stabilizing the compound within the binding cavity of the target protein.

The docked complexes of CP-4 and CP-5 showed contrastingly weak interactions, compared to CP-3 docked complex. Although the formation of H-bond by gly-65 (2.94Å), tyr-505 (2.14Å), and tyr-132 (2.03Å) indicated the polar nature of the CP-4 docked complex. Similarly, the H-bonds visualized in the CP-5 docked complex were present at a distance of 2.10Å for ser-506, and 2.20Å for tyr-118 residue. Moreover, the binding pocket was further stabilized by non-polar hydrophobic contacts.

Lastly, the compound CP-6 also showed strong binding interactions with the amino acid residues of the binding pocket. The formation of H-bond and π-sulfur interactions was suggestive of the stability and polar nature of the mutated docked complex. The conventional H-bonds formed with the hydroxyl groups of the compounds by the following amino acid residues: tyr-505 (1.76Å), his-377 (2.68Å), gly-303 (2.69Å), and tyr-132 (1.96Å) at specific angles of 167.09º, 161.05º, 140.13º, and 165.32º for each respectively. The amino acid residue i-e met-508 was involved in the formation of π-sulfur interaction. Further, various hydrophobic

interactions contributed to stabilizing the binding pocket, facilitating the mode of interaction between the receptor and the ligands.

### 3.4. SIFT analysis

Fig 6 portrays the superimposition of all 6 ligands along with the CCL within the binding cavity of the mutated target protein CYP-51, indicating that all ligands were docked within the defined allosteric site and interacting with the key residues of the binding cavity. Additionally, a fingerprinting analysis has shown that the residue tyr-118, tyr-132, phe-233, met-508, and ser-378, were able to interact over 80% of the ligands docked into the active site. This indicated that these residues were the hotspot residues in the binding cavity the of mutated CYP-51 protein that could help in accommodating potential inhibitors of the protein.

### 3.5. *In silico* validation of ADMET analysis

**3.5.1. Drug-likeness.** Drug-likeness and oral bioavailability are derived from a drug's physicochemical properties. Tables 3 and 4 show the pharmacokinetic and toxicity profiles of the different ligands. In the analysis of the pharmacophore features of the 11,022 compounds to find a potential drug candidate, the six best compounds were screened and subsequently

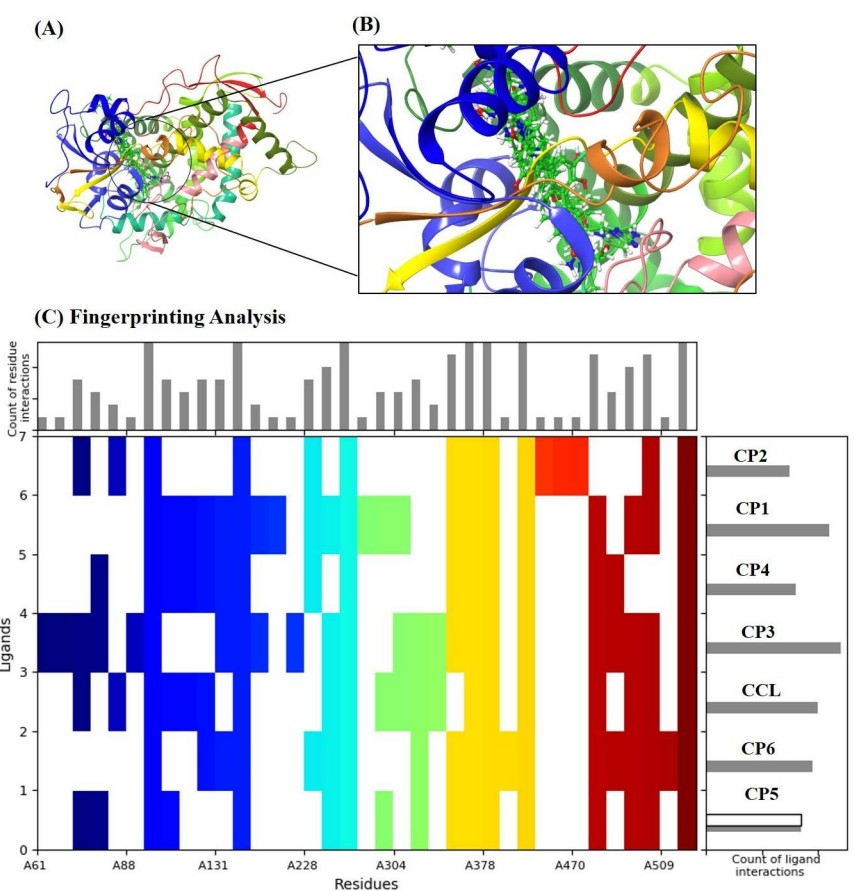

**Fig 6. Interaction fingerprinting analysis illustrating key molecular interactions between selected ligands and the mutated CYP-51 receptor, including hydrogen bonds, hydrophobic contacts, π–π stacking, van der Waals forces, etc.**

**Table 3. Pharmacokinetics and toxicity profiles of novel inhibitors, assessed through SwissADME and pkCSM serves.**

| Compound Codes | Absorption and Distribution | | Metabolism | | Toxicity | |
|---|---|---|---|---|---|---|
| CP-1 | GI | High | CYP1A2 inhibitor | Yes | Hepatotoxicity | No |
| | P-gp substrate | No | CYP2D6 inhibitor | Yes | AMES Toxicity | No |
| | BBB Penetration | Yes | CYP2C19 inhibitor | No | Carcinogenicity | No |
| | Bioavailability | 0.55 | CYP2C9 inhibitor | Yes | hERG | No |
| | Log $S$ (SILICOS-IT) | Poor solubility (−9.32) | CYP3A4 inhibitor | No | | |
| CP-2 | GI | High | CYP1A2 inhibitor | No | Hepatotoxicity | No |
| | P-gp substrate | No | CYP2D6 inhibitor | No | AMES Toxicity | No |
| | BBB Penetration | No | CYP2C19 inhibitor | No | Carcinogenicity | No |
| | Bioavailability | 0.55 | CYP2C9 inhibitor | No | hERG | No |
| | Log $S$ (SILICOS-IT) | Moderate solubility (−5.23) | CYP3A4 inhibitor | No | | |
| CP-3 | GI | Low | CYP1A2 inhibitor | No | Hepatotoxicity | No |
| | P-gp substrate | Yes | CYP2D6 inhibitor | No | AMES Toxicity | No |
| | BBB Penetration | No | CYP2C19 inhibitor | No | Carcinogenicity | No |
| | Bioavailability | 0.55 | CYP2C9 inhibitor | No | hERG | No |
| | Log $S$ (SILICOS-IT) | Moderate solubility (−5.35) | CYP3A4 inhibitor | No | | |
| CP-4 | GI | Low | CYP1A2 inhibitor | Yes | Hepatotoxicity | No |
| | P-gp substrate | No | CYP2D6 inhibitor | No | AMES Toxicity | No |
| | BBB Penetration | No | CYP2C19 inhibitor | No | Carcinogenicity | No |
| | Bioavailability | 0.55 | CYP2C9 inhibitor | No | hERG | No |
| | Log $S$ (SILICOS-IT) | Poor solubility (−7.08) | CYP3A4 inhibitor | No | | |
| CP-5 | GI | High | CYP1A2 inhibitor | No | Hepatotoxicity | No |
| | P-gp substrate | No | CYP2DA inhibitor | Yes | AMES Toxicity | No |
| | BBB Penetration | Yes | CYP2C19 inhibitor | Yes | Carcinogenicity | No |
| | Bioavailability | 0.55 | CYP2C9 inhibitor | Yes | hERG | No |
| | Log $S$ (SILICOS-IT) | Poor solubility (−6.70) | CYP3A4 inhibitor | Yes | | |
| CP-6 | GI | High | CYP1A2 inhibitor | Yes | Hepatotoxicity | No |
| | P-gp substrate | Yes | CYP2DA inhibitor | No | AMES Toxicity | No |
| | BBB Penetration | Yes | CYP2C19 inhibitor | Yes | Carcinogenicity | No |
| | Bioavailability | 0.55 | CYP2C9 inhibitor | No | hERG | No |
| | Log $S$ (SILICOS-IT) | Poor solubility (−7.65) | CYP3A4 inhibitor | No | | |
| R* (posaconazole) | GI | High | CYP1A2 inhibitor | Yes | Hepatotoxicity | No |
| | P-gp substrate | No | CYP2DA inhibitor | Yes | AMES Toxicity | No |
| | BBB Penetration | Yes | CYP2C19 inhibitor | Yes | Carcinogenicity | No |
| | Bioavailability | 0.17 | CYP2C9 inhibitor | Yes | hERG | No |
| | Log $S$ (SILICOS-IT) | | CYP3A4 inhibitor | Yes | | |

subjected to further processing. ADME was used in the early phase of drug design and evaluation to predict the pharmacokinetic and toxicity properties of compounds. To estimate their drug-likeness, absorption, and distribution in body tissues and the brain (so-called ADME properties), the natural compounds were assessed according to criterion parameters. However, it is noteworthy that all the natural compounds had molecular weights less than 500 g/mol, HBA < 10, and HBD < 5; in addition, they did not violate Lipinski's rule of 5, except for the co-crystal ligand, the reference drug (R*) posaconazole.

**3.5.2. Absorption and distribution.** In this study, we used Log P$o/w$ and a descriptor of Log S, which are very lipophilic and readily diffuse across biological membranes. The results

**Table 4. Drug-likeness profiles of novel inhibitors, assessed through SwissADME server.**

| Codes | CP-1 | CP-2 | CP-3 | CP-4 | CP-5 | CP-6 | R* (posaconazole) |
|---|---|---|---|---|---|---|---|
| **Lipinski's Rule** | Yes | Yes | Yes | Yes | Yes | Yes | 2 Violation |
| **HBA** | 1 | 6 | 6 | 3 | 6 | 5 | 9 |
| **HBD** | 4 | 4 | 6 | 5 | 3 | 2 | 1 |
| **MW (g/mol)** | 394.47 | 393.39 | 411.50 | 394.47 | 396.41 | 396.45 | 700.78 |
| **Log P(o/w)** | 1.78 | 1.81 | 1.00 | 3.03 | 2.83 | 3.67 | 3.98 |

HBA: Hydrogen bond acceptor, HBD: Hydrogen bond donor, MW: Molecular weight, Log P: Log of the octanol-water partition coefficient.

of lipophilicity (consensus Log P*o/w*) and water solubility indicated that all the compounds were substantially soluble in aqueous media. This information may serve as guidance for preparing them for pharmaceutical development.

ADME selects Log S as the descriptor to predict solubility. Poorly soluble chemicals show high negative results for Log S. Log S of the compounds CP-3 and CP-5 were the lowest, whereas it was between −10 and 0 for most of the other compounds, i.e., a higher negative value meant higher water solubility; therefore, this distribution might significantly increase the solubility of compounds and affect mutated CYP-51, in terms of the overall Log S.

**3.5.3. Metabolism.** The binary (yes/no) state of primary cytochrome inhibition was expected to be predicted using SwissADME. Given the substantial role of metabolism in drug development, we needed to elucidate its implications for pharmacokinetics, pharmacodynamics, and safety. The ADMET was used for predicting enzyme inhibition (%). The following compounds CP-3, CP-4, and CP-5 did not show inhibitory activity against five CYP P450 isoenzymes and might have affected the ability of the mutated protein. Simultaneously, CP-2, CP-6, CP-7, CP-8, and R* compounds were shown to possess inhibitory effect for CYP-1A2, CYP2C9, CYP2D6, CYP-2C19, and CYP-3A4. Furthermore, CYP2C19 and CYP3A4 were predicted to be substantially inhibited by CP-2 and CP-6. Cytochrome P450 (CYP) 3A, showing the highest abundance in humans, is responsible for metabolizing xenobiotics such as drugs, hormones, carcinogens, and eicosanoids.

**3.5.4. Toxicity.** Toxicity is one of the crucial stages in the design and development of the drug. The results revealed that no toxicity was observed across hepatoxicity, AMES toxicity, carcinogenicity, and hERG in any of the six compounds.

## 3.6. *In silico* validation of DFT properties

The frontier orbitals (HOMO and LUMO) were calculated for the inhibitor CP-3 (4-[2-[4-[(2-aminoethylamino)methyl]phenyl]ethynyl]-N-[(2S)-3-amino-1-(hydroxyamino)-1-oxopropan-2-yl]benzamide;methane). The energies of HOMO and LUMO were related to multiple chemical and pharmaceutical processes. HOMO and LUMO frontier orbitals were also mapped onto the molecular surface as well the HOMO–LUMO gap of CP-3. The HOMO and LUMO values calculated from the electrostatic potential map of the inhibitor would make it much easier to understand where these two frontier orbitals lie in the case of the inhibitors. S3 Table shows the values of the computed quantum chemical descriptors, HOMO and LUMO, for CP-3 in solvation (water) phases. The HOMO value of CP-3 was higher than the LUMO value, which reflected its higher willingness to donate electrons compared to CYP-51. Higher electron acceptor ability was exhibited by CP-3 (4-[2-[4-[(2-aminoethylamino)methyl] phenyl]ethynyl]-N-[(2S)-3-amino-1-(hydroxyamino)-1-oxopropan-2-yl]benzamide; methane), with a comparatively higher LUMO value. These results are indicative of the rapid electron transfer capability of CP-3. In addition, solvation (water phase) dipole moments of

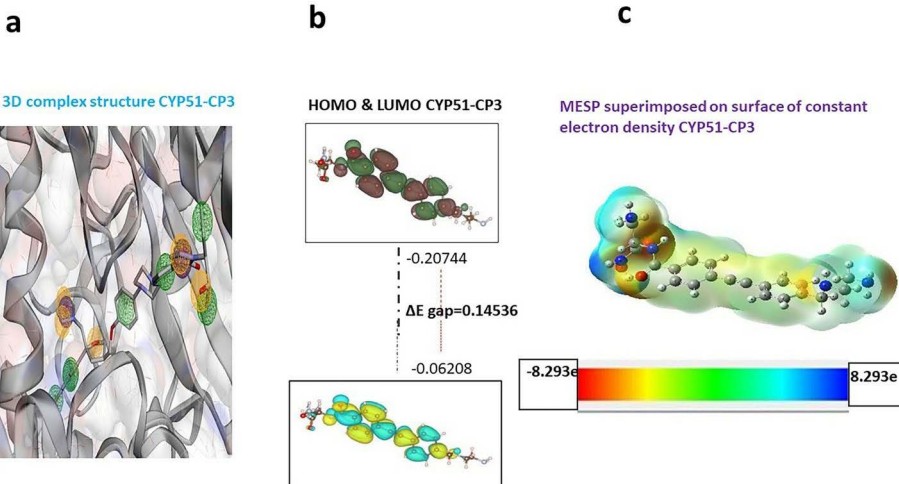

**Fig 7. Visual representation of CP-3 ligand.** (a) 3D conformation of CP-3 bound to CYP-51 within the binding cavity. (b) Optimized molecular representation of FMOs of CP-3 depicting the energy gap between HOMO and LUMO orbitals in atomic units. (c) MESP mapping on the CP-3 ligand indicates a color progression from red (electronegative region) to blue (electropositive region), with green being the neutral region.

CP-3 were in the range of 6.8 to 15.7 Debye (S3 Table). CP-3, as the most active molecule, showed a higher dipole moment (compared to water i-e. 5.569 Debye) (Fig 7).

**3.6.1. Molecular electrostatic potential mapping.** One of the most popular ways of characterizing pharmacologically active compounds through their physical properties and the tendencies of chemical reactivity is MESP mapping. It provides a better understanding of the various properties of interaction with the binding sites of numerous isoform proteins. The ligand's steric and electronic properties were compared to the critical structural pharmacophoric features to understand its binding affinity and selectivity. The changes in the electrostatic potentials throughout the compound were investigated. CP-3 * 4-[2-[4-[(2-aminoethyl amino)methyl]phenyl]ethynyl]-N-[benzamide; methane]. [4-(2S)-3-amino-1-(hydroxyamino)-1-oxopropan-2-yl] was carefully examined, including the ligand–receptor binding selectivity demonstrated by the mutant receptor protein, CYP-51. Based on the data yielded by our modeling results, we also developed the structure-based pharmacophoric features of CP-3 via co-crystal structures or models generated from MD-simulated complexes (discussed below) between the ligand and CYP-51 to unveil the contribution of these activities to bridge–ligand complex formation. The physicochemical properties and electrostatic potentials of the CYP-51 mutant are shown in the presence of CP-3 (Fig 7). The pharmacophore model described the chemical features of the compounds and that the sulfonyl oxygen functional group should be a hydrogen bond acceptor to the side-chain N atoms (Lys-143 and His-468) in CYP-51; however, no such H-bonding interaction was present. Further optimization of a docking solution led CP-3 to form hydrophobic interactions with Pro-88, Val-159, Tyr-161, Glu-126, Ser-353, Phe-357, Leu-428, Leu-429, Ser82, Leu-86, and Met-485. Therefore, it might be regarded as a critical functional feature to discriminate the highly potent and selective CYP-51 inhibitor from those that have been reported so far. CYP -51 binding pocket side chain amino group showed H bond interaction with CP-3. Investigation of the pharmacophore model-defined structure-based chemical features also indicated the involvement of these regions in interacting with key residues such as Lys-143, His-468, Ser-507, and Tyr -505 present at the active site of CYP-51.

MD simulations of CP-3 complexes using the mutated CYP-51 structure were performed in explicit aqueous solution. The RMSD fluctuations of backbone atoms were calculated to examine the dynamic stability of each complex and verify the reasonableness of the results obtained by MD simulations. The RMSD curves of the whole protein and ligands from the binding pocket were obtained for whole images saved during MD simulation with respect to the corresponding starting structure in each complex (Fig 8 ). All systems equilibrated quickly after a few initial fluctuations. Therefore, it was reasonable to calculate each residue's binding free energies, and energy decomposition for ligand binding using a conformation extracted from the simulations. The docked complex of CYP-51 with R* compound (CCL) showed an overall consistent and equilibrated conformation with a minor fluctuation around ~2.5Å (S1 Fig). Contrastingly, the CYP-51–CP-3 complex exhibited a higher fluctuation, with an RMSD of approximately 0.5 Å and ~2.8Å for the protein and ligand, respectively. In this context, R* compound showed relatively stable occupation of the mutated CYP-51's mutual conformation density space while the compound CP-3 showed pronounced fluctuation indicating the flexibility of the docked complex.

### 3.7. *In silico* validation of molecular dynamics simulation

Before applying the virtual screening protocol, MD simulations were carried out using the AMBER software package to verify the binding modes of the ligand and the stability of the docked complexes. Simulations of the lead compound CP-3 with mutated CYP-51 (Y132H) complex were carried out using 200 ns as the top dock scorer and snapshots were taken at various intervals during the simulation trajectory (Fig 9 ).

In addition, RMSF was employed to assess the overall movement of amino acid residues from the protein structure. RMSF was used to average all of the atomic fluctuations for

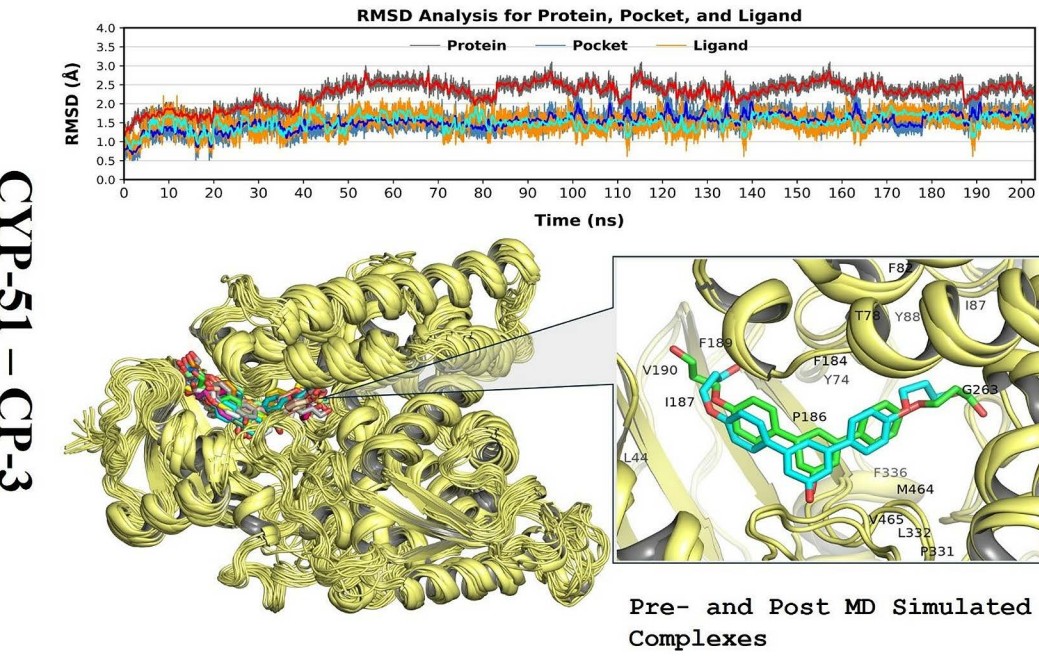

**Fig 8. Molecular dynamics (MD) simulation analysis of the mutated CYP-51 (Y132H) complexes representing time dependent-RMSDs of protein Cα atoms, pocket, and the ligand relative to the protein along with the visual representation of CP-3 docked complex showing ligand binding domain; a pre- and post-MD simulated complexes.**

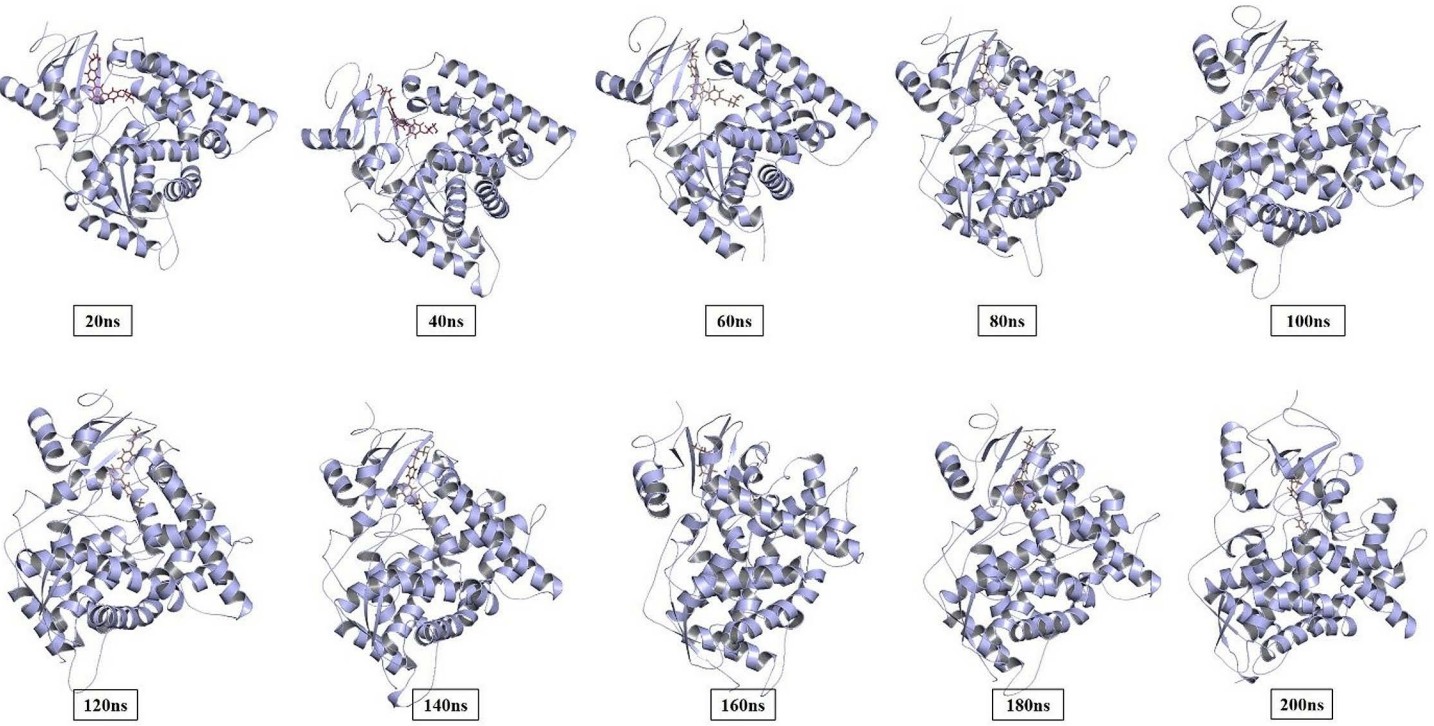

**Fig 9. Snapshots of the trajectory taken at different intervals for the mutated target protein CYP-51.**

individual amino acid residues. The simulation results for the R* compound complexed with mutated CYP-51 protein showed that the RMSF of fluctuated residues ranged up to 4Å. The highest peak was observed for residue ser-378, which was a hotspot interacting residue of the binding pocket (S2 Fig). Conversely, the RMSF plot results for the CP-3 complex showed that the RMSF of fluctuated residues was above 4 Å, which was related to the ligand in this case. The similar RMSF distributions of all protein systems (CYP-51) suggest that CP-3 inhibitors adopted a common binding mode with the CYP-51 mutant. The highest RMSF was determined for the active site of the CYP-51 mutant Y132H, suggesting stable binding by CP-3 and indicating that ser-378 was an important amino acid for potential hydrogen interactions with CYP-51 with a critical role in retention or release within the active site of the target receptor.

The solvent-accessible surface area (SASA) accounts for how much the solvent comes into contact with the protein. In the case of R* compound (CCL), the values ranged from $550Å^2 – 750Å^2$ while for CP-3 ligand, it ranged between $600Å^2$ and $700Å^2$. The plot revealed a steady pattern for SASA whereas, the CP-3 showed an initial rose in SASA, indicating that the protein had grown during early phases, but was constant thereafter.

Finally, the radius of gyration (Rg) determined the folding and unfolding of the secondary structures of the protein, indicating the overall compactness of the protein. A wider Rg fingerprint reflected greater overall flexibility of the biopolymeric system; this was signposted by the extent of the collapse of the protein in our simulation under trapping conditions. Fig 10 shows that both R* compound and CP-3 showed stable and steady conformation ranging between ~22.5 to 22.7, indicating the compactness of protein during the simulation trajectory.

**3.7.1. Binding free energy analysis and per-residue analysis.** Binding energies in the gaseous phase, solvation energy, and entropic contribution were estimated with 1000 snapshots taken every 20 ns for each protein-ligand system using molecular mechanics/

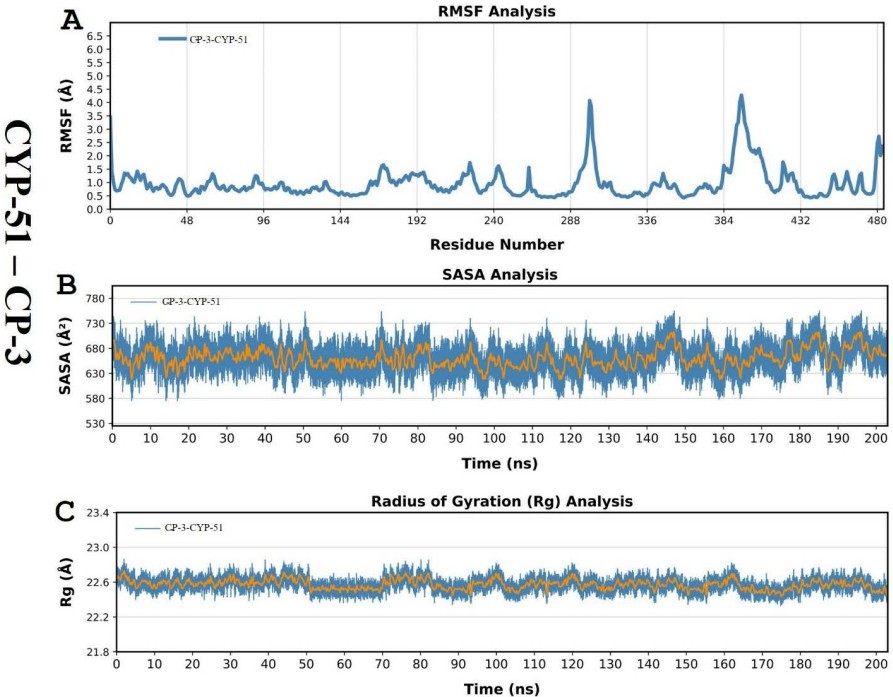

**Fig 10. Graphical representation of RMSF, SASA, and Rg of the lead compound CP-3 complexed with mutated target protein CYP-51 during simulation trajectory.**

Poisson–Boltzmann surface area (MM/PBSA) and MMPBSA.py.MPI1 packages to compare the binding affinities of CP-3 inhibitors against the CYP-51 mutants. The ordering of the DGpred (GB) values, which are binding free energy predictions for CP3 on CYP-51 (−44.83 kcal/mol), and the cohesive effect of these results, together with their experimental affinities, were consistent. In addition to CP-3 binding, we also observed the DGpred (GB/PB) value of the wild-type CYP-51 Y132H mutant. The results indicated that CP-3 binds more tightly to the CYP-51 mutants than to the original, which is consistent with limiting flexibility at the inhibitor-binding site. Using MMGB (PB)SA techniques, the overall binding free energy was further broken down into different binding free energy components to obtain insights into the driving mechanisms behind the selective binding of the ligand CP-3 to the mutant CYP-51. The sum of the electrostatic interaction contributions in vacuum (DGele) and solvent (DGele, sol) disfavored ligand-protein binding, according to the calculated values of individual binding free energy components for complex systems. This indicates that the unfavorable electrostatics of the desolvation contribution oppose the favorable Coulomb interactions between protein and ligand. Although the sum of vdW energy (−54.1774 kcal/mol) and nonpolar solvation-free energies (−7.708 kcal/mol) may be the primary factors influencing the inhibitor's selective binding, the substantial role played by electrostatic components (−20.7448 kcal/mol) in the inhibitor's binding affinity is a favorable contribution to the inhibitor binding to CYP-51 (Fig 11 and Table 5). Conversely, in R* (CCL) the sum of vdW energy and nonpolar solvation-free energies was higher than the CP-3 complex which could be suggestive of strong dispersive forces. Moreover, R* also exhibited a pronounced level of electrostatic interactions, indicating better stabilization (S3 Fig).

The per-residue analysis of the CP-3 docked complex in (S4 Table) revealed that the following residues phe189, met464, leu332, and tyr74 provided strong stability due to the favorable

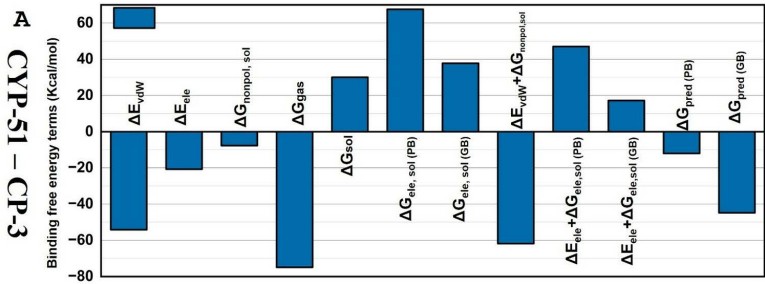

**Fig 11. Plot representing the free binding energies of CYP-51– CP-3 complex during simulation trajectory.**

**Table 5. Calculation of binding free energy of CYP-51– CP-3 complex in comparison to R\* docked complex after simulation trajectory.**

| Binding free energy parameters | CYP-51–CP-3 complex (kcal/mol) | CYP-51-R* (CCL) complex (kcal/mol) |
|---|---|---|
| $\Delta E_{vdW}$ [a] | −54.1447 | −89.8154 |
| $\Delta E_{ele}$ [a] | −20.7448 | −29.2427 |
| $\Delta G_{nonpol, sol}$ [a] | −7.708 | −10.6386 |
| $\Delta G_{gas}$ | −74.8895 | −119.0581 |
| $\Delta G_{sol}$ | 30.055 | 31.4354 |
| $\Delta G_{ele, sol (PB)}$ [a] | 67.5644 | 80.4321 |
| $\Delta G_{ele, sol (GB)}$ [a] | 37.7631 | 42.0741 |
| $\Delta E_{vdW} + \Delta G_{nonpol,sol}$ [a] | −61.8527 | −61.8527 |
| $\Delta E_{ele} + \Delta G_{ele,sol (PB)}$ [a] | 47.0192 | 47.0192 |
| $\Delta E_{ele} + \Delta G_{ele,sol (GB)}$ [a] | 17.2179 | 17.2179 |
| $\Delta G_{pred (PB)}$ [b] | −12.0334 | −46.3383 |
| $\Delta G_{pred (GB)}$ [b] | −44.8345 | −87.6227 |

vdW interactions and non-polar solvation contribution. Additionally, aromatic residues also contributed through hydrophobic and vdW interactions. However, few residues were often slightly destabilized due to unfavorable polar solvation which affects the electrostatic and vdW interactions. Contrastingly, the R\* (CCL) complex showed favorable vdW as well as electrostatic interactions by ile87, pro186, and met464, suggesting them as strong stabilizing residues. Furthermore, the presence of aromatic side chains contributed to stabilizing the complex (S5 Table).

## 4. Discussion

In this study, *C. albicans* proteins were extensively screened using virtual methods and choosing the target was one of the most relevant steps in the search for new ligands with inhibitory potential. Before protein–ligand docking, we screened CYP-51 with ligands using Maestro software and homology-based modeling, which identified six compounds as being computationally useful in downregulating this target [34]. Based on these results, a 3D pharmacophore model was established, which provided reliable results from a search of prospective compounds by identifying similar novel chemical entities in order to screen them for future development of drug candidates [35]. The database was docked with a pharmacophore from

posaconazole using Pharmit. The Pharmit library offers capabilities for defining pharmacophore properties such as HBA and HBD, aromatic regions, hydrophobic regions, negative charge ions, positive charge ions and metal coordination sites. Either of these attributes can be selected manually from the posaconazole structure or automated with Pharmit's tools [36].

A pharmacophore-based virtual screening of the interaction between CYP-51 and posaconazole and other selective drugs was performed. The inhibitors were identified using structure-based virtual screening and docked in the binding pocket of mutant CYP-51, with an inhibitor wild type enzyme interaction (P450 inhibitory parameter [PIP]) score. The interaction between CYP-51 and CP-3(4-[2-[4-[(2-aminoethylamino)methyl]phenyl]ethynyl]-N-[(2S)-3-amino-1-(hydroxyamino)-1-oxopropan-2-yl]benzamide;methane) was much greater than those between CYP-51 and the other screening compounds tested [37]. The CP-3 complex pharmacophores were virtually screened based on CYP-51 (Y132H). The hits from Pharmit pharmacophore-based screening for the docking to the preserved mutant CYP-51 binding pocket were achieved through a series of important steps and results in Maestro Dock. Each hit in the two sets was first docked into multiple conformations to understand how they might bind (in this case, six conformers per hit). In our in-silico analysis, the docking results provided contacts and scores for the docking of each compound to specific residues at the active site of mutant CYP-51. The compound CP-3, with the best docking score (−10.70), showed better binding affinity than other pharmacophore-based screening compounds. The results of the current study also aligned with previous studies emphasizing on inhibitory efficacy of the compounds as an anti-fungal agent against CYP-51 protein [38]. Another study also focused on the potential of the target protein used in the current study, CYP-51, as a promising target for anti-fungal drugs showing remarkable inhibitory activity of the novel compounds against this target [39]. Moreover, one such study further endorsed the inhibitory potential of the target CYP-51 [40]. It should be taken into account that the key residues (ser378, met508,tyr132, etc.) of the binding pocket also formed significantly strong interactions with the compound. CP-3 appears to be the frontline agent because it presents stronger binding affinity and many stable contacts [38].

We show that, with the help of the co-crystal ligand reference (R* drug), better screening against selective compounds, i.e., based on pharmacophores, could be achieved. According to Lipinski's rule, ligands with MW < 500 and HBA < 10 should be high oral absorption substrates. However, the reference drug R* did not conform to Lipinski's rule. Pharmacokinetics considers what a body does to the medicine as opposed to pharmacodynamics, which measures how that drug works inside the body [39]. Ligands were selected based on the pharmacophore's ADME and toxicity profile. Oral bioavailability is the total number of pharmacologically active molecules entering the bloodstream. All selective compounds displayed bioavailability scores of approximately 0.55 compared to the reference drug, which suggests that they had reasonable oral bioavailability according to Lipinski's rule of 5 (LRF) [40]. Several studies also employed a similar approach in investigating inhibitors' potential to be drug-like candidates based on LRF [41,42].

The CYP-51–CP-2 and CYP-51–CP-5 complexes had good LogS values, which indicated increased solubility of the compounds. CP-5 had reduced gastrointestinal absorption potential, which limited its usefulness for oral administration, despite having the best docking score and several important interactions. Other approaches are required to improve its bioavailability. Their high gastrointestinal absorption and good docking score implied that CP-2, CP-3, CP-4, and CP-6 had the potential for successful oral administration; nevertheless, more interactions may improve their therapeutic profile. The increased intestinal absorption of these compounds was expressed as the percentage of the administered dose that passed through the portal vein and showed increased intestinal absorption in the body [43]. It is essential to use

different administration methods or techniques to improve the absorption of CP-3. Because P-glycoprotein (P-gp) pumps drugs out of cells, it can restrict drug absorption and distribution. Its total effectiveness may be impacted by its status as a substrate, especially in the brain and other tissues, and its restricted capacity to pass through the blood-brain barrier (BBB), which lessens its potency when targeting the central nervous system. This may plausibly affect the central nervous system and lower the possibility of drug-drug interactions by not inhibiting the main CYP enzymes (CYP1A2, CYP2D6, CYP2C19, CYP2C9, CYP3A4) [44].

However, the *in silico* analysis investigated that no significant toxic effects in the inhibition of the CYP450 enzyme family were observed, especially for CP-3 and CP-5. Furthermore, the various toxicity endpoints such as mutagenicity, tumorgenicity, reproductive as well as irritating potentials have seemed to have no such toxic effect, and therefore, no potential toxicity risk was associated with the compounds [45]. AMES toxicity, carcinogenicity, and hepatotoxicity were absent, suggesting a good safety profile. Only a moderate portion of the dosage of all virtually screened compounds, as well as the reference drug, was absorbed into the bloodstream. CP-3 had the highest effect and was capable of eliminating large drugs from the body compared to the other ligands, as well as the reference compound [46]. In addition, it had a nontoxic profile. Our results indicate that the HOMO value of CP-3 is larger than the LUMO value, confirming that it has a higher tendency to donate electrons to CYP-51. Moderately active CP-3 had a higher LUMO value, which indicates that it is an electron acceptor with greater abilities. CP-3 is more reactive through fast electron transfer. In another study, dipole moments of CP-3 calculated in the solvation (water) phase varied from 6.8 to 15.7 Debye. In addition, the most active CP-3 had a larger dipole moment (compared to water [5.569 Debye]) [47].

MESP mapping indicated that CP-3, as a selective inhibitor of CYP-51, had unique electronic properties. The highest electronegative potential region for electrophilic attack, as indicated by the deep red color in Fig 7, was located on sulfur and oxygen atoms of the sulfonyl group of CP-3. The sulfonyl oxygen atoms of CP-3 were endowed with an average Mulliken charge of + 1.383 and constituted the most negatively charged region near the molecule. Conversely, another substantially less localized negatively charged region associated with CP-3 was situated above the oxygen atom of its one moiety, which had a Mulliken charge of 0.624. Alternatively, the hydrogen atom bonded to the nitrogen atom of CP-3 was identified as containing a nucleophilic center, as revealed by the cyan color, with a Mulliken charge of 0.424 [48]. The structure-based pharmacophore models derived from the co-crystal structures of CP-3 compounds with CYP-51 or MD-simulated structural models of their complexes with CYP-51 also indicated the participation of these areas in crucial interactions with key residues, such as His-377, Ser-378, and Tyr-132 of CYP-51. Thus, the electrostatic potential features are consistent with the structure-based pharmacophore model generated. The strongest option for MD simulation is CP-3 based on its positive ADMET profile and docking score, especially its modest gastrointestinal absorption and inability to cross the BBB. Moreover, the MD simulations of the complex was stable with an RMSD of 3.5 Å and slight fluctuations, suggesting a robust interaction. According to the RMSD graph for CYP-51 with CP-5, it is possible to use CP-5 because of its low gastrointestinal absorption rate. Although it does not provide an average level of bioavailability to ensure further research, it offers steady binding to the target protein. By applying measures to increase its bioavailability and binding stability, a medicine with a positive effect could be designed. The protein-ligand complex reached a steady state after the initial adjustment time as the system was stabilized with an RMSD of 3.5 Å after minimal fluctuation. Therefore, the binding conformation demonstrated is likely to be a good representation of the way that CP-5 interacts with CYP-51 at a dynamic scale. The low RMSF values obtained indicate that CP-3 should be further studied to develop

a complex with CYP-51 with fewer residues involved in the interaction [49]. When CP-5 is present, the protein is less flexible, indicating that the ligand's influence leads to the retention of the protein's conformational stability, which is advantageous due to the characteristics of the protein's intended use and function. The RMSF study indicated that CP-3 was a suitable compound for further development of complexes, given that it leads to the stabilization of the CYP-51 (Y132H) protein's structure. This is supported by previously described results of MD simulation (53). The phe-233, His-377, Ser-378, and Tyr-64 residues are the most effective in the binding of CP-3 to the CYP-51 protein. The active site of the CYP-51 (Y132H) mutant showed the highest RMSFs compared to all other states [50].

MM/GBSA calculations provide valuable insights into the molecular mechanisms of ligand binding and can aid in the development of novel drugs with enhanced potency and selectivity [51]. More potent inhibitors that target particular protein interactions or areas can be created by identifying which residues contribute the most to the binding energy. This demonstrates CP-3's stable form for binding with the CYP-51 mutant. Therefore, it is likely that vdW (−54.1774 kcal/mol) and nonpolar solvation-free energies (−7.708 kcal/mol) with the highest RMSFs are the key factors responsible for the inhibitors' selective binding. This was seen to a greater extent than the role of electrostatic components (−20.7448 kcal/mol) in the binding. The CP3 compound was identified as a novel inhibitor of *C. albicans.*

## 5. Conclusion

In the present study, we used computational tools, including pharmacophore-based screening and molecular docking, to identify potential inhibitors against mutant CYP-51 *C. albicans.* A pharmacophore-based model was generated and a compound, CP-3, was identified as the lead compound. The compound possessed a promising binding affinity (-10.70 kcal/mol) and interactions with key residues of the targeted mutant protein (CYP-51). Computational results also assessed that the ADMET analysis of CP-3 was found to be potential drug-like candidate for the novel antifungal drug. The DFT analysis also contributed to suggesting the reactive molecular properties of CP-3, as a suitable candidate for the inhibition of resistant fungal strains.

It was noteworthy that the present study was limited to *in silico* analysis employing extensive computational tools and no *in vitro* and *in vivo* experiments had been performed to further validate the inhibitory efficacy of the compound CP-3 against mutated CYP-51. Therefore, future work necessitates broadening the application of the current *in silico* research by further performing experimental validation, as well as chemical modification to enhance the activity, selectivity, and pharmacokinetics of the compound CP-3. It must be taken into account that the key interacting residues such as serine, phenylalanine, etc of the binding cavity should be focused for chemical and structural modification of the compound to further enhance the inhibitory effect. These future perspectives could pave the path for pharmaceutical industries to therapeutically use the lead compound CP-3 as a potential antifungal inhibitor against *C. albicans.*

## Supporting information

**S1 Table. List of novel inhibitors against *Candida albicans* assessed through PubChem along with the 2D and 3D molecular conformations of the compounds prepared using ChemDraw Professional 16.0.**
(DOCX)

**S2 Table. Protein structure-based prediction.**
(DOCX)

**S3 Table. The quality of crystal structure mutated CYP-51 (Y132H) structure.**
(DOCX)

**S4 Table. Physiochemical properties of protein with electrostatic potential of ligand CP-3.**
(DOCX)

**S5 Table. Per-residue scores of lead compound CP-3 complexed with mutated protein CYP-51.**
(DOCX)

**S6 Table. Per-residue scores of R\* compound (CCL) complexed with mutated protein CYP-51.**
(DOCX)

**S1 Fig. Molecular dynamics (MD) simulation analysis of the mutated CYP-51 (Y132H) complexes representing time-dependent-RMSDs of protein Cα atoms, pocket, and the ligand relative to the protein along with the visual representation of R\* (CCL) docked complex showing ligand binding domain.**
(DOCX)

**S2 Fig. Graphical representation of RMSF, SASA, and Rg of the R\* (CCL) complexed with mutated target protein CYP-51 during simulation trajectory.**
(DOCX)

**S3 Fig. Plot representing the free binding energies of CYP-51– R\* (CCL) complex during simulation trajectory.**
(DOCX)

## Author contributions

**Conceptualization:** Saadia Jabeen, Abualgasim Elgaili Abdalla.

**Data curation:** Saadia Jabeen, Hasan Ejaz, Shakeel Waqar, Aisha Farhana, Yasir Alruwaili, Qurban Ali.

**Formal analysis:** Muhammad Umer Khan, Hasan Ejaz, Shakeel Waqar, Aisha Farhana, Muharib Alruwaili, Abualgasim Elgaili Abdalla, Sahar Mudassar.

**Funding acquisition:** Hasan Ejaz, Yasir Alruwaili, Abualgasim Elgaili Abdalla.

**Investigation:** Muhammad Umer Khan, Hasan Ejaz, Aisha Farhana, Muharib Alruwaili, Yasir Alruwaili, Abualgasim Elgaili Abdalla.

**Methodology:** Saadia Jabeen, Muhammad Umer Khan, Hasan Ejaz, Shakeel Waqar, Aisha Farhana, Muharib Alruwaili, Sahar Mudassar, Qurban Ali.

**Project administration:** Hasan Ejaz, Muharib Alruwaili, Abualgasim Elgaili Abdalla.

**Resources:** Hasan Ejaz, Shakeel Waqar, Aisha Farhana, Muharib Alruwaili, Yasir Alruwaili, Sahar Mudassar, Qurban Ali.

**Software:** Saadia Jabeen, Muhammad Umer Khan, Shakeel Waqar, Aisha Farhana, Muharib Alruwaili, Sahar Mudassar, Qurban Ali.

**Supervision:** Muhammad Umer Khan.

**Validation:** Saadia Jabeen, Muhammad Umer Khan, Shakeel Waqar, Yasir Alruwaili, Abualgasim Elgaili Abdalla, Sahar Mudassar.

**Visualization:** Hasan Ejaz, Yasir Alruwaili, Abualgasim Elgaili Abdalla.

**Writing – original draft:** Saadia Jabeen, Yasir Alruwaili.

**Writing – review & editing:** Qurban Ali.

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
