## [Decision Letter · Decision Letter 0]

15 Dec 2024

PONE-D-24-49878Identifying novel inhibitors against drug-resistant mutant CYP-51 Candida albicans: A computational study to combat fungal infectionsPLOS ONE

Dear Dr. Ali,

Thank you for submitting your manuscript to PLOS ONE. After careful consideration, we feel that it has merit but does not fully meet PLOS ONE’s publication criteria as it currently stands. Therefore, we invite you to submit a revised version of the manuscript that addresses the points raised during the review process.

We look forward to receiving your revised manuscript.

Kind regards,

Ahmed A. Al-Karmalawy, PhD

Academic Editor

PLOS ONE

Journal Requirements:

3. We note that your Data Availability Statement is currently as follows: [The data generated or corrected has been given in the manuscript and its supplementary file.] Please confirm at this time whether or not your submission contains all raw data required to replicate the results of your study. Authors must share the “minimal data set” for their submission. PLOS defines the minimal data set to consist of the data required to replicate all study findings reported in the article, as well as related metadata and methods (https://journals.plos.org/plosone/s/data-availability#loc-minimal-data-set-definition). For example, authors should submit the following data: - The values behind the means, standard deviations and other measures reported; - The values used to build graphs; - The points extracted from images for analysis. Authors do not need to submit their entire data set if only a portion of the data was used in the reported study. If your submission does not contain these data, please either upload them as Supporting Information files or deposit them to a stable, public repository and provide us with the relevant URLs, DOIs, or accession numbers. For a list of recommended repositories, please see https://journals.plos.org/plosone/s/recommended-repositories. If there are ethical or legal restrictions on sharing a de-identified data set, please explain them in detail (e.g., data contain potentially sensitive information, data are owned by a third-party organization, etc.) and who has imposed them (e.g., an ethics committee). Please also provide contact information for a data access committee, ethics committee, or other institutional body to which data requests may be sent. If data are owned by a third party, please indicate how others may request data access.

4. Please include captions for your Supporting Information files at the end of your manuscript, and update any in-text citations to match accordingly. Please see our Supporting Information guidelines for more information: http://journals.plos.org/plosone/s/supporting-information .

Reviewers' comments:

Reviewer's Responses to Questions

**Comments to the Author**

1. Is the manuscript technically sound, and do the data support the conclusions?

Reviewer #1: Partly

Reviewer #2: Yes

2. Has the statistical analysis been performed appropriately and rigorously? 

Reviewer #1: N/A

Reviewer #2: Yes

3. Have the authors made all data underlying the findings in their manuscript fully available?

Reviewer #1: Yes

Reviewer #2: Yes

4. Is the manuscript presented in an intelligible fashion and written in standard English?

Reviewer #1: Yes

Reviewer #2: Yes

5. Review Comments to the Author

Reviewer #1: Is the manuscript technically sound, and do the data support the conclusions?

Partly

Has the statistical analysis been performed appropriately and rigorously?

N/A

Have the authors made all data underlying the findings in their manuscript fully available?

Yes

Is the manuscript presented in an intelligible fashion and written in standard English?

Yes

1.Avoid abbreviations in the abstract.

2.Avoid abbreviations in keywords.

3.Please indicate the full name of each abbreviation when it first appears in the draft.

4.Enhance the introduction section of your study by incorporating the specified articles to compare your results with those of other studies.

5.2D structures in Table One should all be in the same size and style.

6.The titles of all figures should be written more scientifically.

7.Use more recent references.

Reviewer #2: The authors of the provided manuscript evaluated the anti-Candida albicans activity of PubChem deposited compounds against the drug-resistant mutant CYP-51 through computational approaches. This manuscript is relevant to the field of drug discovery. Few points are to be addressed prior publication.

1. Authors should elaborate more, providing more details on the target topology and pocket description prior presenting the docking findings.

2. Positive control reference compound should be adopted throughout the study to provide significance for the obtained computational findings in terms of relevant biological translation.

3. Authors should elaborate more on the ligand-target interaction patterns Bonding should be annotated in terms of both the bond distances and angles. Specially for Hydrogen bonding, this type of compound-protein polar interaction should be presented within hydrogen bond distances as well as bond angles since hydrogen bond depend on both. Authors should mention the Hydrogen bond angles as well as their distances, since the strength of hydrogen bonding is based on both parameters in a way to ensure the adequacy of optimum hydrogen bonding.

4. The author provided dissection of the free binding energies calculations estimated from the MD simulation trajectories in terms of ∆G electrostatic, ∆G van der Waal, ∆G polar solvation, ∆G non-polar SASA solvation,…) to identify the dominant energy terms that can guide further ligand optimization and development.

Further, the total free binding energies for compound-target complex should be broken down to the target residue energy contribution to highlight which residues impose the great impact on ligand binding for future targeting and lead optimization

5. Authors are advised to provide snap shoot of the MD simulated ligand-protein complex at different time intervals (example 0ns, 50ns, 100ns, 150ns, and/or 200ns) to track the main conformational/orientation changes for both the compound and surrounding residues over time.

6. Based on the study results, what are the take-away messages. Authors are advised to highlight the suggested structural modifications that would improve the compound’s biological activities based on the in silico findings. These insights would be beneficial for guiding further lead optimization and development.

7. Finally, concerning the conclusion, authors are advised to elaborate more on the future of this work? Will you broaden the scope to another drug target? What are the study limitations and what approaches could be conducted to further address them?

6. PLOS authors have the option to publish the peer review history of their article (what does this mean? ). If published, this will include your full peer review and any attached files.

**Do you want your identity to be public for this peer review?** For information about this choice, including consent withdrawal, please see our Privacy Policy .

Reviewer #1: No

Reviewer #2: **Yes**

---

## [Author Response · Author response to Decision Letter 0]

7 Jan 2025

We sincerely thank you and the reviewers for your detailed and constructive feedback on our manuscript. We have carefully addressed all comments and revised the manuscript accordingly. Below, we provide a detailed point-by-point response to each comment.

Reviewer 1 Comments:

Comment 1: Avoid abbreviations in the abstract.

Response: We appreciate the reviewer's suggestion and have revised the abstract to eliminate abbreviations.

Comment 2: Avoid abbreviations in keywords

Response: We appreciate the respectable reviewer’s recommendation and have accordingly removed all abbreviations from the keywords section.

Comment 3: Please indicate the full name of each abbreviation when it first appears in the draft.

Response: We thank the reviewer for this valuable comment. We have ensured that all abbreviations are fully defined when they first appear in the manuscript.

Comment 4: Enhance the introduction section of your study by incorporating the specified articles to compare your results with those of other studies

Response: We are grateful for this insightful suggestion. The introduction section has been updated to incorporate the specified studies. Furthermore, we have added multiple studies in the discussion section to compare our results with previous studies.

Comment 5: 2D structures in Table One should all be in the same size and style

Response: We express our gratitude for the reviewer's observation. We have standardized the size and style of all 2D structures in Table One to ensure consistency.

Comment 6: The titles of all figures should be written more scientifically

Response: We appreciate the reviewer’s suggestion. The titles of all figures have been revised to adhere to a more scientific and precise format.

Comment 7: Use more recent references

Response: We express our gratitude to the reviewer for highlighting this issue. We have updated the reference list with more recent and relevant studies to enhance the manuscript.

Reviewer 2 Comments:

Comment 1: Authors should elaborate more, providing more details on the target topology and pocket description prior to presenting the docking findings.

Response: We thank the reviewer for your valuable suggestion. We have revised the manuscript to include a more detailed description of the target protein in the result section, ensuring a clearer context before presenting the docking results.

Comment 2: Positive control reference compound should be adopted throughout the study to provide significance for the obtained computational findings in terms of relevant biological translation.

Response: We appreciate the esteemed reviewer’s constructive feedback. The manuscript has been revised to incorporate an analysis of the standard reference compound, highlighting its significance across all investigated parameters. It should be taken into account that the molecular docking and ADMET profiling all-encompassed that analysis for the reference compound. Additionally, the comparative analysis was further incorporated in the MD simulations, presented in supplementary files.

Comment 3: Authors should elaborate more on the ligand-target interaction patterns Bonding should be annotated in terms of both the bond distances and angles. Specially for Hydrogen bonding, this type of compound-protein polar interaction should be presented within hydrogen bond distances as well as bond angles since hydrogen bond depend on both. Authors should mention the Hydrogen bond angles as well as their distances, since the strength of hydrogen bonding is based on both parameters in a way to ensure the adequacy of optimum hydrogen bonding.

Response: We express our gratitude to the esteemed reviewer for this technical observation. We have conducted a revised binding interaction analysis for all compounds, including the reference compound, incorporating detailed annotations of bond distances and angles for each interaction. Particular emphasis has been placed on hydrogen bonds to ensure an accurate representation of their strength and adequacy.

Comment 4: The author provided dissection of the free binding energies calculations estimated from the MD simulation trajectories in terms of ∆G electrostatic, ∆G van der Waal, ∆G polar solvation, ∆G non-polar SASA solvation,…) to identify the dominant energy terms that can guide further ligand optimization and development.

Further, the total free binding energies for compound-target complex should be broken down to the target residue energy contribution to highlight which residues impose the great impact on ligand binding for future targeting and lead optimization.

Response: We sincerely appreciate your insightful suggestion. We have added target residue energy contribution in the result section, highlighting the impact of residues on ligand binding.

Comment 5: Authors are advised to provide snap shoot of the MD simulated ligand-protein complex at different time intervals (example 0ns, 50ns, 100ns, 150ns, and/or 200ns) to track the main conformational/orientation changes for both the compound and surrounding residues over time.

Response: We thank the reviewer for this excellent suggestion. We have added snapshots of the MD-simulated complexes at specified time intervals in the result section to visualize conformational and orientation changes of the ligand and surrounding residues over time.

Comment 6: Based on the study results, what are the take-away messages. Authors are advised to highlight the suggested structural modifications that would improve the compound’s biological activities based on the in silico findings. These insights would be beneficial for guiding further lead optimization and development.

Response: We appreciate the suggestion; we have improvised and discussed in the conclusion section by focusing on the key residues for structural and chemical modification of the compound to enhance the inhibitory efficacy of the compound for therapeutic purposes.

Comment 7: Finally, concerning the conclusion, authors are advised to elaborate more on the future of this work? Will you broaden the scope to another drug target? What are the study limitations and what approaches could be conducted to further address them?

Response: We are grateful for the thoughtful insights. We have expanded the conclusion to discuss the potential future directions of this research, including broadening the scope to other drug targets. Additionally, we have outlined the study’s limitations and proposed approaches to address them in subsequent investigations.

---

## [Decision Letter · Decision Letter 1]

20 Jan 2025

Identifying novel inhibitors against drug-resistant mutant CYP-51 Candida albicans: A computational study to combat fungal infections

PONE-D-24-49878R1

Dear Dr. Ali,

We’re pleased to inform you that your manuscript has been judged scientifically suitable for publication and will be formally accepted for publication once it meets all outstanding technical requirements.

Kind regards,

Ahmed A. Al-Karmalawy, PhD

Academic Editor

PLOS ONE

Reviewers' comments:

Reviewer's Responses to Questions

**Comments to the Author**

1. If the authors have adequately addressed your comments raised in a previous round of review and you feel that this manuscript is now acceptable for publication, you may indicate that here to bypass the “Comments to the Author” section, enter your conflict of interest statement in the “Confidential to Editor” section, and submit your "Accept" recommendation.

Reviewer #1: All comments have been addressed

Reviewer #2: (No Response)

2. Is the manuscript technically sound, and do the data support the conclusions?

Reviewer #1: Yes

Reviewer #2: (No Response)

3. Has the statistical analysis been performed appropriately and rigorously? 

Reviewer #1: N/A

Reviewer #2: (No Response)

4. Have the authors made all data underlying the findings in their manuscript fully available?

Reviewer #1: Yes

Reviewer #2: (No Response)

5. Is the manuscript presented in an intelligible fashion and written in standard English?

Reviewer #1: Yes

Reviewer #2: (No Response)

6. Review Comments to the Author

Reviewer #1: All the comments have been addressed

Is the manuscript technically sound, and do the data support the conclusions

yes

Has the statistical analysis been performed appropriately and rigorously?

N/A

Have the authors made all data underlying the findings in their manuscript fully available?

yes

Is the manuscript presented in an intelligible fashion and written in standard English?

yes

Reviewer #2: (No Response)

7. PLOS authors have the option to publish the peer review history of their article (what does this mean? ). If published, this will include your full peer review and any attached files.

**Do you want your identity to be public for this peer review?** For information about this choice, including consent withdrawal, please see our Privacy Policy .

Reviewer #1: No

Reviewer #2: **Yes**

---

## [Editor Report · Acceptance letter]

PONE-D-24-49878R1

PLOS ONE

Dear Dr. Ali,

I'm pleased to inform you that your manuscript has been deemed suitable for publication in PLOS ONE. Congratulations! Your manuscript is now being handed over to our production team.

Kind regards,

on behalf of

Associate Professor Ahmed A. Al-Karmalawy

Academic Editor

PLOS ONE